REGISTERED REPORT PROTOCOL

# Efficacy and safety of the early implementation of a multimodal rehabilitation program in mechanically ventilated patients: A randomized clinical trial protocol

Jorge Iván Alvarado Sánchez[1,2ʘ]*, Laura Maria Castillo Morales[1ʘ],
Yenny Rocio Cardenas Bolivar[1], Valentina Montañez Nariño[1],
Maria Valentina Stozitzky Ríos[1], Catherine Lissell Arévalo Guerrero[3],
Miguel Leonardo Pulido Bobadilla[3], Diana Marcela Melo Rojas[3],
Ana Gabriela López Rubio[4], Diana Carolina Ortíz Moreno[4],
Paula Andrea Barreto Garzón[3], Marisol Murillo[3], Andrés Felipe Mora-Salamanca[1]

1 Department of Critical Medicine and Intensive Care, Fundación Santa Fe de Bogotá, Bogotá, Colombia,
2 Department of Physiology, Faculty of Medicine, Universidad Nacional de Colombia, Bogotá, Colombia,
3 Department of Physical Medicine & Rehabilitation, Fundación Santa Fe de Bogotá, Bogotá, Colombia,
4 Department of Pulmonary Diseases, Fundación Santa Fe de Bogotá, Bogotá, Colombia

ʘ These authors contributed equally to this work
* jialvarados@unal.edu.co

## Abstract

### Background

Intensive care unit-acquired weakness (ICUAW) and post-intensive care syndrome (PICS) are significant complications among critically ill patients, leading to prolonged hospital stays, increased healthcare costs, and reduced quality of life. Early physical therapy in the ICU has shown promise in mitigating these adverse outcomes, yet randomized controlled trials (RCTs) evaluating the combined effects of physical, respiratory, occupational, and speech therapy initiated within the first 24–72 hours are lacking.

### Methods

This single center, controlled clinical trial aims to establish the efficacy and safety of an early multimodal rehabilitation program (MRP) compared with a late MRP in mechanically ventilated patients admitted to the ICU. Adult patients (≥18 years) with a Barthel score ≥70 and requiring invasive mechanical ventilation for more than 24 hours will be included. Participants will be randomly assigned to early MRP, initiated within 24 hours post-intubation, or late MRP, starting 72 hours post-intubation. The primary outcome is the duration of mechanical ventilation. Safety will be assessed by comparing the number of adverse events between groups. The MRP includes passive and active interventions from physical, speech, respiratory, and occupational therapy teams. Therapy intensity and

**Data availability statement:** No datasets were generated or analysed during the current study. All relevant data from this study will be made available upon study completion

**Funding:** The author(s) received no specific funding for this work.

**Competing interests:** The authors have declared that no competing interests exist.

type are adjusted according to the patient's Richmond Agitation-Sedation Scale (RASS) score. Patients with lower RASS scores (indicating deeper sedation) will receive primarily passive interventions (e.g., passive range-of-motion exercises), while those with higher RASS scores (indicating lighter sedation or alertness) will engage in more active therapies (e.g., active-assisted exercises and functional mobility activities). For patients with RASS ≥2, occupational therapy will focus on behavioral and environmental modulation, employing calming techniques, sensory stimuli, and cognitive tasks, with family education to support spatiotemporal orientation. This adaptive approach ensures patient safety and optimizes therapy based on the patient's current sedation level and clinical condition.

## Dissemination

The results will be published in peer-reviewed journals and presented at conferences.

## Trial registration

ClinicalTrials.gov NCT06133504. Registered on November 11, 2023.

## Introduction

Post-intensive care syndrome (PICS) is a significant complication that affects a large proportion of critically ill patients. PICS encompasses a range of physical, cognitive, and mental health impairments that persist long after discharge from the intensive care unit (ICU) [1]. These complications contribute to prolonged hospital stays, increased healthcare costs, and reduced quality of life for survivors of critical illnesses [1,2]. Since these long-term complications are significant, reducing the ICU length of stay could ameliorate their incidence. Patients requiring mechanical ventilation often-present prolonged ICU stays compared to non-mechanically ventilated patients admitted in the ICU. Therefore, mechanical ventilation is a major driver of poor outcomes and a key target for early interventions [3].

Early physical therapy in the ICU has been shown to be a crucial intervention to mitigate these adverse outcomes [4–8]. By initiating mobilization and physical rehabilitation soon after ICU admission, studies have demonstrated improvements in muscle strength, a reduction in the duration of mechanical ventilation, and overall enhancement of functional recovery [4–8]. Early physical therapy reduces the deleterious effects of immobility, thus preventing the rapid onset of muscle atrophy and accelerating the weaning from mechanical support. Consequently, the implementation of early physical therapy protocols is recognized as a vital component of ICU care [4–8].

Despite the established benefits of early physical therapy, there is a lack of randomized controlled trials (RCTs) evaluating the combined effects of physical, respiratory, occupational, and speech therapies within the first 24–72 hours after ICU admission. To our knowledge, no existing RCT has comprehensively assessed the

synergistic impact of these four therapies initiated in this critical early period. Addressing this gap is essential for developing evidence-based protocols aimed at improving outcomes for ICU patients.

## Methods: Participants, intervention and outcomes

The study protocol was prepared following the Standard Protocol Items: Recommendations for Interventional Trials (SPIRIT) [9]. The SPIRIT 2013 checklist and the full protocol (Spanish and its English translation) and can be consulted as S1, S2 and S3 Files, respectively. A summary of the schedule of enrolment, interventions and assessments can be consulted as Fig 1.

### Research ethics approval

This study has been designed and will be conducted in accordance with the Helsinki Declaration [10], the Belmont Report [11], and the Colombian legislation (Resolución No. 8430 de 1993) [12]. The Ethics Committee at the Fundación Santa Fe de Bogotá approved this study on April 24, 2024 (CCEI-16320–2024). Protocols and amendments will be submitted to the Ethics Committee. Once approved, we will update the clinicaltrials.org registration (NCT06133504) and submit the amendments to this journal. Similarly, amendments will be communicated to researchers, clinical staff, and study participants.

The Ethics Committee audits the study twice a year, with progress reports submitted every three months. The interim analyses will be conducted by the Institutional Clinical Studies Office (in Spanish, Subdirección de Estudios Clínicos), which will provide a report to be submitted to the Ethics Committee. Based on the ethics committee's analysis, the trial may be suspended or terminated, considering protocol deviations and adverse events (AE). Nonserious (AE) will be summarized and reported monthly, and the main investigator will immediately report all serious AE to the Ethics Committee within 24 hours of identifying the event.

ICU physicians, therapists, and nursing staff will be trained to obtain informed consent from eligible study participants or their legal representatives. Written informed consent will be obtained after an ICU physician or an ICU research team member has verified the inclusion and exclusion criteria and explained the study's purpose, intervention, duration, benefits, risks, confidentiality measures, and contact information to the participants or their legal representatives. The ICU physician will also clarify that participation is voluntary and will not affect the quality of their medical care if they choose not to participate or withdraw their consent at any time during the study. No compensation will be provided to the study participants; however, patients experiencing AE or medical complications in any group will receive complete medical assistance. Written informed consent (in Spanish and English) can be found in S4 and S5 Files.

Personal data collected from study participants will be protected under confidentiality and personal data processing clauses stated in Colombian legislation (Ley No. 1581 de 2012) [13]. The collected data will be recorded and stored in REDCap, with exclusive access granted to the study researchers. The data will be anonymized to prevent identification of individuals. Third parties (data analysts, outcome assessors, and research auditors, among others) will only have access to anonymized datasets. However, the final trial dataset will be exclusively available to the ICU research team. After trial completion, the datasets and all related documentation will be stored for 20 years as required by Colombian legislation (Resolución No. 2378 de 2008) [14].

### Study design and setting

A single center controlled clinical trial will be conducted with two groups of patients. The experimental group, also known as the "early multimodal rehabilitation program (MRP) arm", will start the MRP in the first 24 hours after intubation. The control group, also known as the "late MRP" group, will start MRP 72 hours after intubation (Fig 2). Beyond the MRP starting time, clinical treatment will not vary between the groups.

| | STUDY PERIOD | | | | | | |
|---|---|---|---|---|---|---|---|
| | Enrollment | Allocation | Post-allocation | | | | Close-out |
| TIMEPOINT** | $-t_1$ | $0$ | $t_1$ | $t_2$ | $t_3$ | $t_4$ | $t_5$ |
| **ENROLLMENT:** | | | | | | | |
| **Eligibility screen** | X | | | | | | |
| **Informed consent** | X | | | | | | |
| **Allocation** | | X | | | | | |
| **INTERVENTIONS:** | | | | | | | |
| *Early MRP* | | | ←———————————→ | | | | |
| *Late MRP* | | | ←———————————→ | | | | |
| **ASSESSMENTS:** | | | | | | | |
| *Ventilation parameters* | X | X | X | X | X | | |
| *Functional measurements* | | | | | X | X | |
| *Time from intubation to extubation* | | | | | X | | |
| *Noninvasive Ventilation* | | | | | X | X | |
| *Extubation failure* | | | | | X | | |
| *Delirium* | | | | | X | X | |
| *Days of MV, sedation, and HS* | | | | | | X | |
| *Adverse events* | | | | | X | X | |
| *Mortality* | | | | | X | X | X |

**Fig 1. SPIRIT schedule of enrolment, interventions and assessments.** HS: hospital stay; MRP: multimodal rehabilitation program; MS: mechanical ventilation. *Group allocation is based on the timing of interventions: Early MRP interventions start within the first 24 hours post-intubation, while Late MRP interventions begin after 72 hours. The frequency and duration of interventions are standardized across both groups: Speech therapy consists of 2 sessions of 10 minutes per day; Physical and Respiratory therapy involve 2 sessions of 30 minutes per day; and Occupational therapy includes 1 session of 30 minutes per day.

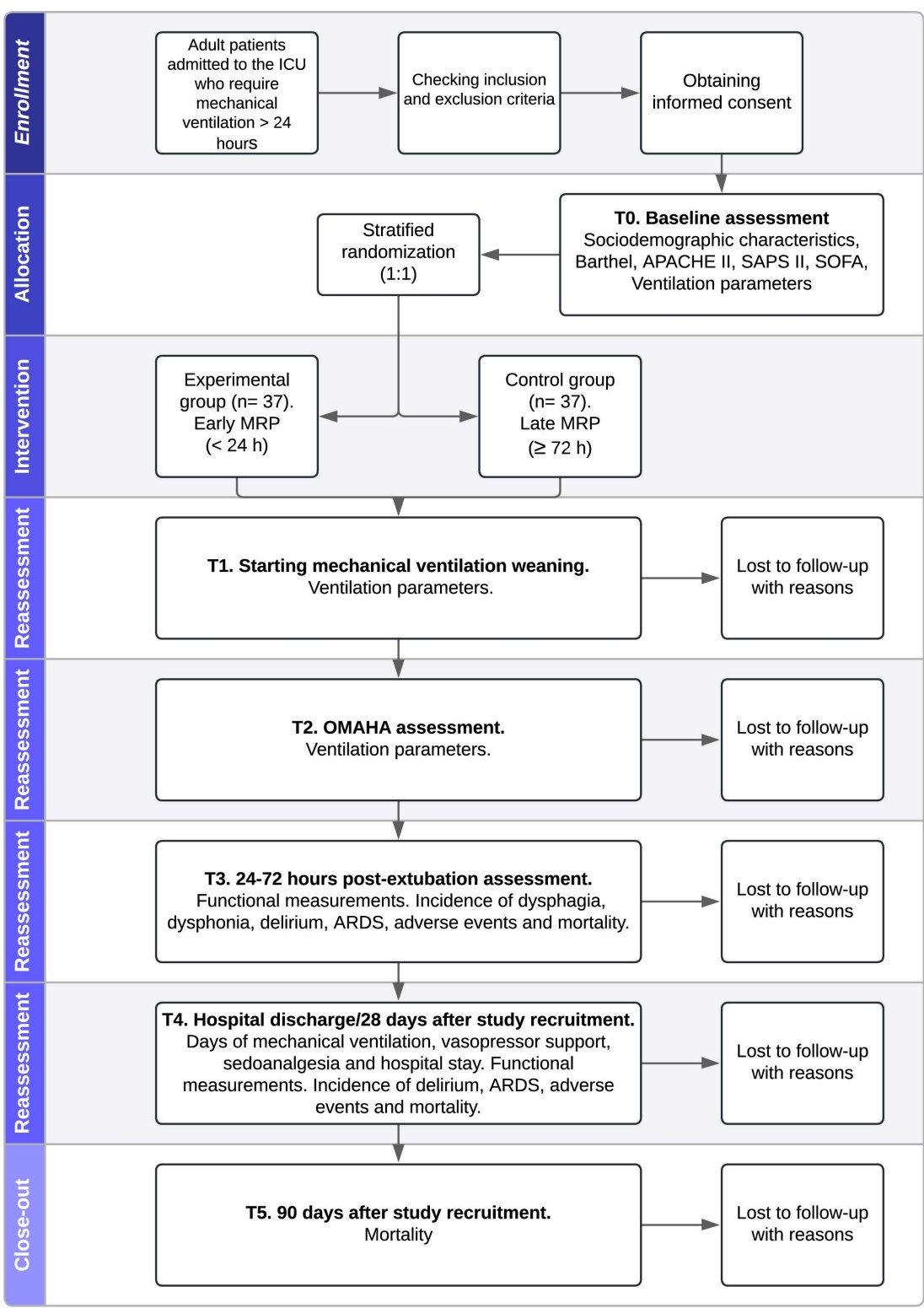

**Fig 2. Flow diagram of the study.** APACHE II: Acute Physiology and Chronic Health disease Classification System II; ARDS: acute respiratory distress syndrome; ICU: intensive care unit; MRP: multimodal rehabilitation program; SAPS II: Simplified Acute Physiology Score II; SOFA: Sequential Organ Failure Assessment Score; T: time point.

## Objective

Our main aim is to establish the efficacy and safety of early MRP compared with late MRP in mechanically ventilated patients admitted to the ICU. The main outcome is the difference in the duration of mechanical ventilation between the two groups. Safety will be assessed by comparing the number of serious and nonserious AE between groups. Additional secondary and exploratory efficacy outcomes will be measured and compared among groups.

## Eligibility criteria

**Inclusion criteria.** Adult patients (≥18 years) with a Barthel score ≥70 admitted to the ICU who required invasive mechanical ventilation through an endotracheal tube for a period longer than 24 hours will be included. A Barthel score of ≥70 ensures that participants have a reasonable level of functional independence before ICU admission, allowing for accurate assessment of the impact of the rehabilitation intervention on post-ICU recovery. Requiring invasive mechanical ventilation for more than 24 hours targets a population with a sufficient duration of mechanical support to benefit from early rehabilitation interventions.

Exclusion criteria

• Invasive mechanical ventilation through tracheostomy or a nasotracheal tube.

• History of head and neck surgery at any time prior to ICU admission.

• Patients directly admitted to the ICU due to cardiac arrest by any cause.

• Airway burn.

• Burns involving ≥ 50% of the body surface area.

• Chronic obstructive pulmonary disease (these patients usually require prolonged mechanical ventilation due to the underlying respiratory dysfunction [15])

• Patients referred from another institution.

• Demyelinating diseases or neuromuscular junction disorders at ICU admission.

• Patients requiring neuromuscular blockade.

• Patients with a life expectancy of ≤180 days will be excluded based on their primary diagnosis (e.g., advanced cancer, end-stage heart failure), comorbidities, recent clinical course, medical judgment by the ICU team, and functional status as assessed by the Barthel index.

• ICU readmissions.

• Participation in other rehabilitation clinical trials.

• Liver or kidney transplant.

• Patients with an active cancer diagnosis undergoing oncological treatment (chemotherapy/surgery) at the time of eligibility assessment.

## Interventions

The early MRP is defined as passive and active therapeutic interventions performed by physical, speech, respiratory, and occupational therapy teams, starting within the first 24 hours after orotracheal intubation. The Late MRP comprises the same interventions, but begins 72 hours after orotracheal intubation. These interventions are part of the standard care provided to all ICU patients and are routinely performed in our institution.

Each therapeutic intervention will be standardized based on the discipline (physical, speech, respiratory, or occupational therapy), and the patient's condition and tolerance. The specific intervention for each session will be adjusted according to the Richmond Agitation-Sedation Scale (RASS) score and functional assessments performed by the respective therapists.

## RASS-Driven Protocol

The MRP interventions will be conducted following a RASS-driven protocol to optimize the daily therapeutic input and minimize downtime due to sedation or agitation. All therapy disciplines (physical, speech, respiratory, and occupational therapy) will provide interventions according to the level of sedation, with the type and intensity of therapy adapted based on the most recent RASS score prior to the commencement of treatment (Table 1).

Min: Minutes; RASS: Richmond Agitation-Sedation Scale.

Oxygenation Criteria for Intervention:

• Fraction of inspired oxygen (FiO2) ≤ 0.6.

• Peripheral oxygen saturation (SpO2) ≥ 90%.

• Respiratory rate ≤ 30 breaths per minute.

• Positive end-expiratory pressure (PEEP) ≤ 10 cmH2O.

**Table 1. Description of the multimodal rehabilitation program interventions.**

| RASS score | | Therapeutic discipline | | | |
|---|---|---|---|---|---|
| | | Physical | Speech | Respiratory | Occupational |
| ≤ -2 | At this deep sedation level, patients will receive passive and passive-assisted interventions | Passive mobilization, muscle recruitment, joint mobilization, and reflex inhibition will be performed. The aim is to maintain joint mobility and prevent muscle atrophy. | Passive proprioceptive stimulation (e.g., craniofacial alignment, jaw and tongue mobilization) and orofacial muscle stimulation will be conducted. | Suctioning of secretions, passive diaphragmatic training, and bronchial hygiene techniques will be provided to maintain pulmonary function. | Multisensory stimulation (tactile, proprioceptive, vestibular, gustatory, auditory, olfactory) will be applied to maintain sensory input and prevent cognitive decline. |
| -1 to +1 | Patients who are more alert will engage in passive-assisted and active interventions | Active motor activation, bed mobility exercises, transfers, walking exercises, and electrostimulation for large muscle groups and respiratory muscles. The goal is to enhance functional mobility and strength. | Active neuromuscular stimulation, hydration of the oral mucosa, and exercises targeting supra- and infrahyoid muscles will be implemented. | Directed coughing, diaphragmatic exercises, incentive spirometry, and non-invasive respiratory muscle training will be applied to improve respiratory function and facilitate weaning from mechanical ventilation. | Active participation in daily tasks, focusing on sensory and cognitive tasks to enhance autonomy in functional movements such as reaching, grasping, and releasing. |
| ≥ 2 | At this level, most therapies will remain the same as in RASS -1 to +1, but occupational therapy will focus on behavioral modulation | | | | Behavioral and environmental modulation will focus on calming techniques and reducing agitation through sensory stimuli and cognitive tasks, with family education for spatiotemporal orientation. |
| Session duration (min) | | 30 | 10 | 30 | 30 |
| Daily frecuency (number of sessions per day) | | 2 | 2 | 2 | 1 |

## Monitoring

All staff involved will be trained to accurately assess the RASS score and other scales used in the study to reduce variability. Interventions will be documented consistently, and ICU monitoring or medical interventions that do not interfere with the study protocol will not be a cause for exclusion.

## Participant withdrawal

Study participants will be withdrawn from the study in the following scenarios:

- Adverse events: According to the Common Terminology Criteria for Adverse Events (CTCAE) [16], AE will be systematically recorded and categorized as follows:

  ◦ Grade 1: Mild; asymptomatic or mild symptoms; clinical or diagnostic observations only; intervention not indicated.

  ◦ Grade 2: Moderate; minimal, local or noninvasive intervention indicated; limiting instrumental activities of daily living (e.g., preparing meals, shopping for groceries or clothes, using the telephone, managing money).

  ◦ Grade 3: Severe or medically significant but not immediately life threatening; hospitalization or prolongation of hospitalization indicated; disabling; limiting self-care activities of daily living (e.g., bathing, dressing and undressing, feeding self, using the toilet, taking medications, and not bedridden).

  ◦ Grade 4: Life-threatening consequences; urgent intervention indicated.

  ◦ Grade 5: Death related to AE.

The decision to withdraw participants will be made to prioritize patient safety and prevent potential harm from study-specific interventions. All major AE will be rapidly notified to the Ethics Committee (<24 hours), who will determine if the AE was associated to the intervention. Although minor AE will also be notified to the Ethics Committee (<one week), the principal investigator will discussed with the attending physician to determine whether the AE is directly related to the study intervention. If the intervention is associated with the AE, the participant could be withdrawn according to the AE classification, the likelihood of reoccurring and the willingness of the participant to continue in the clinical trial. Patients withdrawn due to complications will continue to receive full ICU care as part of standard treatment. For those still on mechanical ventilation at the time of withdrawal, their time on the ventilator will be recorded up to the point of withdrawal. This data will be included in the analysis as censored observations in the intention-to-treat analysis.

- Health complications: If a participant develops a health condition *de novo* that might affect their participation in the study (e.g., acute myocardial infarction, pulmonary embolism, intracerebral hemorrhage, cardiac dysfunction requiring inotropic or vasopressor support), they will be withdrawn from the study.

- Revocation of informed consent: Participants and their legal representatives have the right to revoke previously given consent to participate in this trial at any time. Medical attention will not have any repercussions.

- Early discontinuation of the trial: If early evidence indicates that early MRP is significantly more or less beneficial than late MRP in invasive mechanically ventilated patients, the study will be stopped, and the best MRP strategy will be offered to all patients. Similarly, if early evidence suggests that early MRP or late MRP is harmful, the study will be reviewed by an Independent Data Monitoring and Safety Committee. Based on their recommendations, the study may be terminated to ensure patient safety.

- Transference to another healthcare institution:

  ◦ Data from patients transferred to another facility will be retained and if appropriate, it will be included in the intention to treat analysis. If the proportion of missing data due to the transfer is less than 20%, we will use data imputation

                                                

methods to account for the missing values. This ensures that the contribution of these patients is not lost and that the sample size and statistical power are preserved. Transfers will be recorded as a separate variable, and we will explore their potential impact on the study outcomes through sensitivity analyses.

## Adherence strategies

We will implement educational strategies, audits, feedback, and reminders for all ICU staff. Educational strategies will focus on training nurses, physicians, and therapists on the informed consent process, study participant eligibility, and the interventions involved. A pilot study will be conducted to identify challenges in implementing the study protocol and to ensure comprehension of the processes and study forms. Based on the pilot study findings, we may adjust the initial forms and data collection tools.

Furthermore, we will monitor adherence to the MRP intervention among therapy staff via an electronic reporting system, which will also track RASS scores, vital signs before and after the intervention, session duration, and any adverse events. The reporting system will work as follows:

- Daily logs will be maintained by the clinical team, recording the frequency, duration, and type of interventions received by each study participant.

- A weekly review of these logs will be conducted by a monitoring committee to ensure adherence to the intervention protocol.

- Supervisors or study coordinators will periodically audit the therapeutic interventions to verify compliance with the study protocol.

  Participants will be classified as adherent or non-adherent to the intervention according to the following criteria:

- Missed interventions: In case, the number of sessions missed exceeds more than 30% of the scheduled sessions during the ICU stay (due to health complications, scheduling conflicts, among other reasons) the patient will be classified as non-adherent.

- Session duration: Therapeutic sessions lasting less than 50% of the expected time (i.e., less than 15 minutes per session) are considered missing sessions. If this pattern repeats in more than 30% of scheduled sessions during the ICU, the patient will be classified as non-adherent.

- Incorrect interventions: Interventions applied incorrectly (e.g., applying active interventions to patients having a RASS score$<$-2) are considered missing sessions and follow the missing session criteria to classify patients as adherent or non-adherent.

The results will be communicated to the staff, and we will provide both general and individualized feedback while also gathering their feedback on the protocol implementation. If needed, additional training on specific topics will be offered to both the clinical staff and the research team.

Additionally, study reminders will be placed in the participant's room and the nurse station. Clinical staff will also have access to a 24-hour phone line for any trial-related questions.

## Deviations from the study protocol

◦ External Factors: Deviations due to unintentional or external factors, such as ICU procedures (e.g., central line insertions), surgeries, cardiac arrest during ICU stay, staffing limitations, or uncontrolled sedation practices, will not lead to participant withdrawal. These deviations will be documented as protocol deviations and addressed in sensitivity analyses to evaluate their potential impact on study outcomes

- Minor protocol deviations (an accidental or unintentional change to, or non-compliance with the Ethics Committee approved procedures without Ethics Committee prior approval that does not increase the risk or decrease the benefit to the patient nor significantly affects the subject's rights, safety or welfare and/or the integrity of the research data [17]) will be documented and reported to the Ethics Committee. These cases will be addressed in a sensitivity analysis to determine their potential impact on the study outcomes.

- Major protocol deviations (an accidental or unintentional change to, or non-compliance with the Ethics Committee approved procedures without Ethics Committee prior approval that increase the risk or decrease the benefit to the patient or significantly affects the subject's rights, safety or welfare and/or the integrity of the research data [17]) will be evaluated by the Ethics Committee, who will determine if the study can continue or must be terminated.

Documentation and Reporting: All non-adherent cases will be documented and reported. The research team will analyze minor deviations and, accordingly, general or particular feedback will be given to researchers, clinical team or study participants. Deviations caused by external factors will be explicitly categorized and incorporated into sensitivity analyses to maintain transparency and study integrity

## Outcomes

We will collect the following baseline variables at the time of randomization (within the first 24 hours after intubation) to ensure comparability between groups: sociodemographic data (e.g., age, sex), clinical scores (Barthel, Sequential Organ Failure Assessment [SOFA] Score, Simplified Acute Physiology Score II [SAPS II], Acute Physiology and Chronic Health Disease Classification System II [APACHE II]), main diagnosis, body mass index, and ventilator parameters (tidal volume, respiratory rate, positive end-expiratory pressure, plateau pressure, airway resistance, and the ratio of arterial oxygen partial pressure [PaO2 in mmHg] to fractional inspired oxygen [PaO2/FiO2 ratio]).

Ventilator parameters (e.g., tidal volume, respiratory rate, PEEP, $FiO_2$, plateau pressure, airway resistance, and $PaO_2/FiO_2$ ratio) will be recorded at baseline (within the first 24 hours after intubation) and daily thereafter, ideally before the morning physiotherapy session.

## Primary outcome

Our main outcomes are the difference in the duration of invasive mechanical ventilation, defined as the time (days) from orotracheal intubation to complete orothracheal extubation, between the experimental and control groups. We selected this outcome since it directly compares the efficacy of early MRP to that of late MRP for diminishing the time to extubation.

## Safety outcomes

Concerning safety, we will measure the following adverse events and compare their occurrence between groups:

- Altered blood pressure: Decrease in systolic arterial pressure >30% or mean arterial pressure >15%.

- Cardiac arrhythmia leading to hemodynamic instability.

- Oxygen desaturation: Oxygen saturation <80% for at least 10 minutes.

- Gastrointestinal complications: nausea, vomiting or diarrhea.

- Agitation: RASS > 2 requiring sedation.

- Removal of invasive line.

- Altered neurological status: Decrease in Glasgow Coma Scale (GCS) <8.

- Accidental extubation.

- Healthcare-associated pneumonia.

- Bronchoaspiration.

**Secondary outcomes**

We will also measure differences between the experimental and control groups regarding the following outcomes at T4 (up to day 28 after intubation in case the patient remains hospitalized or up to hospital discharge, in case the hospitalization period lasted less than 28 days):

- Number of ventilator-free days (up to 28 days).

- Number of days with delirium (up to 28 days).

- Number of days with delirium in the ICU (up to 28 days).

- Number of days with delirium in the general ward (up to 28 days).

- Number of days under sedation (up to 28 days).

- Number of days without sedation (up to 28 days).

- Extubation failure.

- Need for noninvasive mechanical ventilation.

**Exploratory outcomes**

We will explore the following outcomes:

- Time from intubation to starting the MRP

- Muscle strength 24 hours postextubation.

- The highest level of mobility 24 hours postextubation.

- Grip strength 24 hours postextubation.

- Dysphagia 72 hours postextubation.

- Dysphonia 72 hours postextubation.

- The time from extubation to restarting oral intake.

- Requirement of noninvasive mechanical ventilation postextubation.

- Number of days hospitalized.

- Number of days in the ICU.

- Barthel score 28 days postintubation/hospital discharge.

- Cognitive function 28 days postintubation/hospital discharge.

- Activities of daily life assessment 28 days postintubation/hospital discharge.

- Rapid sensory and dexterity test 28 days postintubation/hospital discharge.

- Incidence of all-cause mortality up to 90 days postintubation.

- **Ventilator Parameters Trend:** defined as daily percentage change is calculated as the difference between the parameter value (e.g., $FiO_2$, PEEP, tidal volume) on the current day and the previous day, divided by the previous day's value, then multiplied by 100.

- **Ventilator Parameters Trend:** The daily percentage change in ventilator parameters (e.g., $FiO_2$, PEEP, tidal volume) is calculated as the difference between the value on the current day and the previous day, divided by the previous day's value, and then multiplied by 100.

- **Ventilatory Stability Days:** Stability is defined as the absolute change in ventilator parameters between consecutive days being less than 10% of the value on the previous day. The total number of consecutive days meeting this condition will be counted.

- **Weaning Progression:** Progression is defined as a reduction in ventilator support that meets specific thresholds, such as a decrease in $FiO_2$ by 10% or more, a reduction in pressure support by at least 2 $cmH_2O$, or an improvement in the $PaO_2/FiO_2$ ratio by 20% or more compared to the previous day.

### Participant timeline

The data will be collected at six time points: before allocation, at the beginning of mechanical ventilation weaning, during the OMAHA+ assessment, 24–72 days postextubation, at hospital discharge or on the 28th day after orotracheal intubation, and 90 days after intubation (Figs 1–2). The list of variables and their descriptions and categorizations can be found in S2 and S3 Files.
    Mechanical ventilation weaning will start when the following pre-requisite are accomplished:

- The indication to start mechanical ventilation has been successfully addressed.

- Patient is hemodynamically stable, or the vasopressor/inotropic support is diminishing (in case the patient is on vasopressor/inotropic support).

- The awakening has started (diminishing sedative medications)

- FIO2<40% and SpO2>89%

    The OMAHA+ assessment is an institutional checklist used in our ICU to identify patients, currently in the weaning of mechanical ventilation support process, who can be successfully extubated. It includes the Yang and Tobin index, along with additional clinical, hemodynamic, and ventilatory parameters that reduce the risk of extubating failure and reintubation. To initiate the extubating process, a patient must meet at least 90% of the criteria on the OMAHA+ checklist. A full description of the OMAHA+ assessment can be found in S6 File.

### Sample size

To estimate the sample size, we included previous parameters published by Schweickert and colleagues [18]. They reported that the mean duration of mechanical ventilation in patients receiving usual physical therapy was 6.1 (±1.4) days compared to 3.4 (±1.25) days in a group of patients receiving early therapy. Given an alpha error of 0.05 and a beta error of 0.9, we estimated the sample size according to the following formula:

$$n = \frac{2\left(\frac{Z_\alpha}{2} + Z\beta\right)^2 \times \left(SD1^2 + SD2^2\right)}{ES^2}$$

where $n$ is the sample size, $\frac{Z\alpha}{2}$ is the critical value of the normal distribution at α/2 (2.81), $Z\beta$ is the Zβ is the Z score associated with the beta level (1.28), $SD1$ is the standard deviation in the group receiving usual physical therapy, and $SD2$ is the standard deviation in the group receiving early physical therapy [18]. The effect size (ES) was estimated as follows:

$$ES = \frac{|\mu1 - \mu2|}{\sqrt{\frac{SD1^2 - SD2^2}{2}}}$$

where $\mu1$ is the mean duration of mechanical ventilation in the group receiving usual physical therapy, $\mu2$ is the mean duration in the group receiving early physical therapy, and $SD1$ and $SD2$ are the same standard deviations used in the first equation. By substituting the values into the second equation, the ES equals to 2.03.

$$ES = \frac{|6.1 - 3.4|}{\sqrt{\frac{1.40^2 - 1.25^2}{2}}} = 2.03$$

Therefore, the resulting equation obtained by replacing the values in the first equation is:

$$n = \frac{2\,(2.81 + 1.28)^2\ \times\ (1.40^2 +\ 1.25^2)}{2.03^2} = 30.29\ \sim\ 31\ patients$$

We estimated 31 patients per intervention group (62 patients in total). An additional 20% (6 patients per group) were added to minimize the loss of statistical power in the case of patients withdrawing from the study or missing data (74 total study participants).

## Methods: Assignment of interventions

### Recruitment and allocation

In 2023, more than 600 mechanically ventilated patients were admitted to our ICU. Consequently, we anticipate recruiting approximately 15–20 patients per month, accounting for exclusion criteria and the willingness of legal representatives or patients to participate in the study.

Study participants will be randomly assigned to each group using a stratified randomization strategy to ensure balance by age (<65 and ≥65 years), sex (male and female), and comorbidities (Charlson comorbidity index <8 and ≥8) resulting in eight strata. The allocation sequence will be generated through a computer-generated random numbers system, a function provided by REDCap. Study participants will be irreversible assigned by REDCap to either the intervention group or the control group once eligibility is confirmed. The randomization process will be monitored and documented electronically within REDCap, ensuring transparency and minimizing the potential for bias. Additionally, REDCap will maintain a log of all randomization events, which can be audited if necessary.

Given the nature of our study, which involves applying an MRP, it is not feasible to blind the intervention to either the medical staff or the patients. However, to minimize bias, the assessment of the primary outcome will be performed by an independent evaluator who is blinded to the group assignments of the participants. This evaluator will be responsible for assessing the outcome without knowledge of which intervention was applied. The statistical analysis will be conducted by an independent researcher who will have access to the data but will not be involved in the outcome assessment.

## Methods: Data collection, management, and analysis

### Data collection and management

The data will be collected at six time points (Fig 1). The case report form (CRF) does not include any personal data, and every study participant will be given a randomized identification number to anonymize the data. The CRF includes demographic and clinical variables (information from T0) and information already registered in the institutional electronic medical records system. Information from time points 1 and 2 will be recorded by a respiratory therapist. Each therapist, together with the ICU research team, will record data at time points 3 and 4. Finally, an ICU research team member will contact the patients through a phone call or their relatives to verify their health status 90 days after intubation (T5).

Both the clinical and research teams will receive training on the proper completion of the case report form (CRF) and other study forms, which can be consulted as S7 File. Clear procedures will be established to ensure consistent data collection among the teams. The following strategies will be implemented:

- Data Verification: Two members of the clinical team will verify any extreme or outlier values identified between T0 and T4 to ensure accuracy and determine whether these values reflect true clinical outcomes or potential data entry errors.

- Electronic CRF: An electronic CRF and other study forms will be used to capture hours and dates related to each specific study time point, facilitating the monitoring of adherence to the study protocol.

If a patient provides their informed consent and wishes to withdraw from the study, we will inquire about their reasons for discontinuing participation. Similarly, if a patient is withdrawn or deviates from the study protocol for any reason, this information will be summarized and published in the final report and scientific paper. Nonetheless, we will perform an intention-to-treat analysis and a sensitivity analysis that includes patient data up until the time point where they revoked consent or deviated from the protocol. All patients who experience serious adverse events will be included in the analysis.

### Data management

Study data will be recorded in an electronic CRF on the REDCap platform (https://www.project-redcap.org/) using the randomized identification number assigned to each study participant. Relevant information from the institutional electronic medical records system will be extracted and entered into the electronic CRF.

To ensure data quality and reliability, we will implement the following strategies:

- Data Entry Checking: Verification of consistency, incongruities, reliability, outliers, and missing CRF data at the time of data entry in REDCap.

- Double Data Entry: Two independent researchers will enter the data, and the resulting databases will be compared to identify any discrepancies.

### Statistical analysis

All statistical analyses will be conducted by data analysts blinded to group allocation to ensure unbiased results. The analyses will follow the intention-to-treat (ITT) principle and will be performed using R software.

The normality of quantitative variables will be assessed with the Shapiro-Wilk test. Normally distributed variables will be presented as means with standard deviations, while non-normally distributed variables will be reported as medians with interquartile ranges. Qualitative variables will be expressed as frequencies and proportions.

Comparisons between groups will use appropriate tests depending on data type and distribution:

- Continuous variables: T-test or Mann-Whitney U test.

- Categorical variables: $\chi^2$ test or Fisher's exact test.

**Primary outcome analysis.** The primary outcome, duration of mechanical ventilation, will be analyzed as follows:

1. **Comparison of Means:** A T-test (or Mann-Whitney U test for non-normal distributions) will compare the means between groups.

2. **Survival Analysis:** Kaplan-Meier estimates and Cox regression models will account for the time-dependent nature of the outcome. Patients still ventilated at the end of the follow-up period (up to 28 days) will be treated as censored observations. Differences between groups will be assessed using a log-rank test.

For patients withdrawn while still ventilated, their duration on ventilation will also be treated as a censored observation to maintain adherence to the ITT principle.
**Secondary and subgroup analyses.**

1. **Ventilator Parameters Analysis:** Ventilator parameters (e.g., tidal volume, respiratory rate, PEEP, $FiO_2$, plateau pressure, airway resistance, $PaO_2/FiO_2$ ratio) will be recorded daily, ideally before the physiotherapy session. Baseline parameters will be compared between groups using T-tests or Mann-Whitney U tests. Daily data will be analyzed to identify trends over time and relationships with the intervention using repeated measures or mixed-effects models.

2. **Subgroup Analyses:** Patients with ICU stays shorter than 72 hours will be compared with those with longer stays to explore differences in outcomes. Sensitivity analyses will evaluate the effect of including or excluding patients with shorter ICU stays or those withdrawn due to complications.

**Multivariable analyses.** Factors influencing the duration of mechanical ventilation will be explored through univariable Cox regression models. Variables with a p-value <0.25 or clinical relevance (e.g., age, sex, delirium) will be included in multivariable models, refined using a backward stepwise approach based on Akaike (AIC) and Bayesian (BIC) information criteria. The linearity of continuous variables will be verified using Martingale residual plots, with stratification applied for non-linear relationships.

**Confounding and interaction assessment.** Potential confounders will be evaluated by secondary regression models. Variables causing a change >15%-20% in adjusted estimates will be retained as confounders. Interaction effects will be assessed by introducing interaction terms into regression models, with significance determined at $p < 0.05$.

**Extreme values.** Regression models will be used to identify leverage points that could disproportionately influence results. Extreme values that do not alter the model will remain in the analysis, as they may represent clinically relevant findings.

**AE analysis.** Descriptive statistics will summarize the frequency and distribution of AE across different grades. Statistical comparisons between groups will be performed to evaluate differences.

**Sensitivity analyses.** Sensitivity analyses will evaluate the impact of patient withdrawals and protocol deviations on study outcomes. If no statistically significant differences are observed, a post-hoc power analysis will determine whether the sample size was sufficient to detect meaningful differences.

**Logistic regression models.** Logistic regression will analyze dichotomous outcomes such as failed extubation, post-extubation non-invasive ventilation, tracheostomy, prolonged mechanical ventilation, and dysphonia. Assumptions of logistic regression, including linearity in the log-odds, independence of errors, and absence of multicollinearity, will be verified through residual diagnostics.

## Dissemination

### Dissemination

We are committed to the complete and transparent publication of all study results, whether positive or negative. Once the data are analyzed, the results will be published in a peer-reviewed journal and presented at both Colombian and international events.

We will include as authors every individual who meets the criteria and recommendations proposed by McNutt and colleagues [19]. Once the manuscript is prepared, it will be submitted to American Journal Experts for grammar, style, and spell checking. No additional editing services or artificial intelligence will be used for manuscript enhancement.

### Trial status

We have not yet recruited the first patient. We plan to start in April 2025 and expect to finish in April 2026. The data will be stored in our institutional Redcap platform and will be accessible upon reasonable request.

## Supporting information

**S1 Checklist. SPIRIT 2013 Checklist.**
(PDF)

**S2 File. Protocol. The Spanish protocol was approved by the research ethics committee** .
(PDF)

**S3 Protocol. The study protocol translated to English.**
(PDF)

**S4 File. Informed Consent in Spanish.**
(PDF)

**S5 File Informed Consent in English** .
(PDF)

**S6 File. Checklist. OMAHA+ Assessment Checklist** .
(PDF)

**S7 File. CRF. Case report form** .
(PDF)

## Author contributions

**Conceptualization:** Jorge Iván Alvarado Sánchez, Laura Maria Castillo Morales, Yenny Rocio Cardenas Bolivar, Valentina Montañez Nariño, Maria Valentina Stozitzky Ríos, Catherine Lissell Arévalo Guerrero, Miguel Leonardo Pulido Bobadilla, Diana Marcela Melo Rojas, Ana Gabriela López Rubio, Diana Carolina Ortíz Moreno, Paula Andrea Barreto Garzón, Marisol Murillo, Andrés Felipe Mora-Salamanca.

**Data curation:** Jorge Iván Alvarado Sánchez, Andrés Felipe Mora-Salamanca.

**Formal analysis:** Jorge Iván Alvarado Sánchez, Andrés Felipe Mora-Salamanca.

**Funding acquisition:** Jorge Iván Alvarado Sánchez, Andrés Felipe Mora-Salamanca.

**Investigation:** Jorge Iván Alvarado Sánchez, Laura Maria Castillo Morales, Yenny Rocio Cardenas Bolivar, Valentina Montañez Nariño, Maria Valentina Stozitzky Ríos, Miguel Leonardo Pulido Bobadilla, Diana Marcela Melo Rojas, Ana

Gabriela López Rubio, Diana Carolina Ortíz Moreno, Paula Andrea Barreto Garzón, Marisol Murillo, Andrés Felipe Mora-Salamanca.

**Methodology:** Jorge Iván Alvarado Sánchez, Laura Maria Castillo Morales, Yenny Rocio Cardenas Bolivar, Valentina Montañez Nariño, Maria Valentina Stozitzky Ríos, Catherine Lissell Arévalo Guerrero, Miguel Leonardo Pulido Bobadilla, Diana Marcela Melo Rojas, Ana Gabriela López Rubio, Diana Carolina Ortíz Moreno, Paula Andrea Barreto Garzón, Marisol Murillo, Andrés Felipe Mora-Salamanca.

**Project administration:** Jorge Iván Alvarado Sánchez, Catherine Lissell Arévalo Guerrero, Andrés Felipe Mora-Salamanca.

**Resources:** Jorge Iván Alvarado Sánchez, Andrés Felipe Mora-Salamanca.

**Software:** Jorge Iván Alvarado Sánchez, Andrés Felipe Mora-Salamanca.

**Supervision:** Jorge Iván Alvarado Sánchez, Andrés Felipe Mora-Salamanca.

**Validation:** Jorge Iván Alvarado Sánchez, Andrés Felipe Mora-Salamanca.

**Visualization:** Jorge Iván Alvarado Sánchez, Andrés Felipe Mora-Salamanca.

**Writing – original draft:** Jorge Iván Alvarado Sánchez, Andrés Felipe Mora-Salamanca.

**Writing – review & editing:** Jorge Iván Alvarado Sánchez, Andrés Felipe Mora-Salamanca.

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
