## [Decision Letter · Decision Letter 0]

3 Sep 2024

PONE-D-24-31135Efficacy and Safety of the Early Implementation of a Multimodal Rehabilitation Program in Mechanically Ventilated Patients: A Randomized Clinical Trial ProtocolPLOS ONE

Dear Dr. Alvarado Sánchez,

Thank you for submitting your manuscript to PLOS ONE. After careful consideration, we feel that it has merit but does not fully meet PLOS ONE’s publication criteria as it currently stands. Therefore, we invite you to submit a revised version of the manuscript that addresses the points raised during the review process.  The issues appear to be mostly statistical in nature as well as with the study design.

We look forward to receiving your revised manuscript.

Kind regards,

Steven E. Wolf, MD

Academic Editor

PLOS ONE

Journal Requirements:

Reviewers' comments:

Reviewer's Responses to Questions

**Comments to the Author**

1. Does the manuscript provide a valid rationale for the proposed study, with clearly identified and justified research questions?

Reviewer #1: Yes

Reviewer #2: Partly

2. Is the protocol technically sound and planned in a manner that will lead to a meaningful outcome and allow testing the stated hypotheses?

Reviewer #1: Partly

Reviewer #2: Partly

3. Is the methodology feasible and described in sufficient detail to allow the work to be replicable?

Reviewer #1: Yes

Reviewer #2: No

4. Have the authors described where all data underlying the findings will be made available when the study is complete?

Reviewer #1: Yes

Reviewer #2: Yes

5. Is the manuscript presented in an intelligible fashion and written in standard English?

Reviewer #1: Yes

Reviewer #2: Yes

6. Review Comments to the Author

You may also provide optional suggestions and comments to authors that they might find helpful in planning their study.

Reviewer #1: There are some major statistical problems in this protocol:

1. A survival analysis is given as the data analysis technique for the primary outcome, yet the sample size is based on a t-test. The test of equality of survival curves must use sample size formulas based on survival data (e.g., Rosenberger and Lachin, Randomization in Clinical Trials, Wiley, 2016, Section 2.6.3).

2. The randomization procedure ("computer generated") is not given, nor a discussion of potential stratification, blocking, etc. (e.g., Rosenberger and Lachin, Randomization in Clinical Trials, Wiley, 2016, Chapter 3).

3. Sealed envelopes were used in the 1948 streptomycin trial, but these days we use computer generated assignments available through Redcap or on a Shiny app.

4. The data analysts obviously cannot be blinded to the data! How can they do comparisons and test the primary outcome if they are blinded? What needs to be blinded is the assessment of the primary outcome, which should be done independently of those providing the intervention.

5. Very complicated set of interventions, but if successful, the results will be interesting.

Reviewer #2: Manuscript PONE-D-24-31135

Title: Efficacy and Safety of the Early Implementation of a Multimodal Rehabilitation Program in Mechanically Ventilated Patients: A Randomized Clinical Trial Protocol

Thank you to the authors for a well-articulated and referenced manuscript. The study protocol requires additional methodological detail and consideration of the most appropriate outcome measure for this study. The manuscript will be publishable with moderate adjustments, particularly to the Introduction and Methods, in this reviewer’s view.

The fundamental question for the authors in the study design is to consider whether starting the input two days (48hrs) apart is enough to create a measurable study effect, and whether the sample groups are large enough to observe this effect. For instance, tailoring interventions to the RASS score could result in participants from both groups receiving similar and, or low volumes of therapy because heavy sedation or agitation excludes them for two days of their stay from receiving ‘active’ treatments.

Please adjust the manuscript and, or respond to the comments below.

Abstract:

1. Please make it clearer for the reader as to how will the RASS score affect therapy input.

Introduction:

2. The study justification as articulated in the Introduction needs rework. The primary outcome measure is not post-ICU outcomes as purported as the main reason for early intervention, it is time on the ventilator. Why would the investigators not use a primary measure that is a post-ICU physical or quality of life outcome, as justified in by the Introduction?

Methods:

3. Eligibility Criteria – requires additional detail and justification.

4. Will patients with ICU length of stay shorter than 72 hours be included? How will this be accounted for in the study and statistical analyses? How might this impact the efficacy of group allocation (randomisation)?

5. Line 181 - Will patients with isolated smoke inhalation injury (airway burn) be excluded? If so, why?

6. Line 183 - Why are patients with COPD excluded?

7. Line 187 - How will life expectancy criteria be applied at recruitment?

8. Line 204 - Currently the daily number of treatments is not standardised. This reviewer strongly suggests that the number of intervention events be standardised to one or two per day per discipline to reduce study variability.

9. Similarly, the volume of treatment (minutes of input) applied per participant is a significant variable in this study. How will the investigators account for the daily differences?

10. Line 212 - Is there a better way to schedule the interventions to reduce variation in active treatment volume per patient? For instance, rather than accept that therapy is not completed or sessions markedly shortened due to periods of high sedation and agitation, could the investigators apply a RASS driven protocol which optimises the daily input (minimises down time) for the patient through the standardising treatment according to the strengths of different disciplines of the team?

11. Line 299 – How will adverse events be managed in the study analyses?

12. Line 308 – Will the patient withdrawn due to complications no longer receive treatment in ICU? Is there any specific justification for why those with complications are withdrawn? Complications are relatively common in ICU and strictly applying this criteria may impact the sample size, feasibility of the study and ability to measure secondary outcomes with integrity.

13. Line 314 – This criterion is troublesome as the treatment is not standardised. How will non-compliance be confirmed and at what ‘amount’ of deviation from the protocol will constitute need for withdrawal? More detail needed please.

14. Line 328 – How will transfers to another facility be treated? Will their data prior to transfer be included in the study?

Outcomes

15. Line 356 – how often will ventilator parameters be collected and how will they be included in the Statistical Analyses?

16. The primary outcome is not justified by the Introduction as it focusses on post-ICU acquired weakness, PICS, physical and quality of life outcomes. It would seem more likely that the primary outcome has been chosen for convenience and not related to the substance of the study and planned interventions?

17. Line 371 – cardiac arrythmias are mentioned as a reason for withdrawal earlier. How will adverse events be delineated as an occurrence to be recorded, or result in participant withdrawal which will affect the sample size and power of the trial analyses?

18. Line 384 – The criteria 28 days after intubation or hospital discharge is ambiguous. Please clarify.

Participant Timeline

19. Line 414 – is there a typo – 24-72 hours post-extubation?

Sample Size

20. Line 438 – 440 – is there a typo related to the group description for SD1 and SD2?

21. The primary outcome is measured in hours, however the sample size has been calculated using days. Does this make a difference to the mathematics of the calculation?

22. The sample size seems particularly low for an ICU study. There are a multitude of variables which could affect the time ventilated (including the patient diagnosis or reason for critical care), the standardisation of the interventions and cause withdrawal of a patient according to the proposed criteria. Has the mathematics of this sample size calculation been applied to account for so many variables which may not be managed effectively by stratification of random allocation? Please revisit.

Recruitment and Allocation

23. Stratification for age has been applied. Why not also for diagnosis?

Data collection and management

24. It is unclear how long a patient needs to remain in the study before their case is considered as part of the sample. For instance, with short ICU length of stay, will there be adequate time for the interventions to be applied and impact the outcome?

25. Line 501 – Data verification – this statement reads like a statistical ‘get out of jail free’ card. Why would extreme values be assessed for verification if not to exclude them from the study analysis? Please clarify the intent of this action in the study.

26. Line 508 - ? typo (tense) – please check expression.

Statistical Analysis

27. Exactly which variables will be carried forward using the intention-to-treat principles?

28. Line 544 - Please explain the risk variable. If the patient is not extubated, how can the time of mechanical ventilation be used in the proposed variable? How is this model interpreted? Also, as time on the ventilator is the primary variable, how can it be included as a co-variate in the same statistical model?

29. The statistical modelling is complex and involves not only known variables but also interaction terms and ad hoc tests. Is the relatively small sample size with repeated measures longitudinal design likely to support an interpretable analysis for any of the secondary outcomes?

Figures

30. Figure 1 – The interventions subsection doesn’t add value or needs adjustment to demonstrate that the group allocation (time and volume of interventions) is the key variable in the study.

Thank you for developing this study.

7. PLOS authors have the option to publish the peer review history of their article (what does this mean?). If published, this will include your full peer review and any attached files.

Reviewer #1: No

Reviewer #2: **Yes: **Dale Wesley Edgar

---

## [Author Response · Author response to Decision Letter 1]

16 Oct 2024

Letter to reviewers

Dear Editor,

Thank you for allowing us to send you the revised version of our study protocol. Below, you will find the responses to each of your requirements.

Journal Requirements:

We have reviewed the manuscript and supplemental material and adjusted to the PLOS ONE’s style requirements.

Thank you for your valuable feedback. In response to your request regarding the Data Availability Statement, we have updated it as follows:

• Data Availability Statement: All data generated or analyzed during this study will be available upon study completion and reasonable request. If the data cannot be made publicly accessible, this will be clearly stated along with the reasons. Currently, all relevant data are included in the manuscript or supporting information files.

We hope this revision meets the requirements, and we remain available for any further clarification.

Thank you for your feedback. We have made the necessary adjustments, and the ethics statement now appears only in the Methods section of the manuscript, as requested.

Reviewers' comments:

Reviewer's Responses to Questions

Comments to the Author

Reviewer #1: There are some major statistical problems in this protocol:

1. A survival analysis is given as the data analysis technique for the primary outcome, yet the sample size is based on a t-test. The test of equality of survival curves must use sample size formulas based on survival data (e.g., Rosenberger and Lachin, Randomization in Clinical Trials, Wiley, 2016, Section 2.6.3).

Thank you for your insightful feedback. We would like to clarify that our primary outcome, the duration of invasive mechanical ventilation, was conceptualized and analyzed as a quantitative variable in units of time (days), not as a time-to-event variable typically used in survival analysis. For this reason, we did not consider the survival analysis into the sample size estimation. We used a difference in means approach, which we deemed more appropriate for this outcome.

In the revised manuscript, we have clarified this distinction to ensure that the variable is correctly understood as quantitative (days), and not a time-to-event outcome (lines 366-370). Although our outcome could be perceived as time-dependent, we felt that analyzing it as a difference in means better aligned with the nature of the data and our research objectives.

We acknowledge your suggestion regarding the use of survival data formulas and will carefully consider this point in future analyses or similar studies

2. The randomization procedure ("computer generated") is not given, nor a discussion of potential stratification, blocking, etc. (e.g., Rosenberger and Lachin, Randomization in Clinical Trials, Wiley, 2016, Chapter 3).

Thank you for your valuable feedback. In response to your suggestion regarding the randomization procedure, in the protocol we included a stratified randomization strategy. Specifically, participants will be stratified by age (<65 and ≥65 years), sex (male and female) and comorbidities (Charlson comorbidity index <8 and ≥8) resulting in eight strata. We added his information in the manuscript (lines 493-496).

3. Sealed envelopes were used in the 1948 streptomycin trial, but these days we use computer generated assignments available through Redcap or on a Shiny app.

Thank you for your valuable suggestion regarding the randomization procedure. We changed the sealed envelopes to a random allocation sequence provided by REDCap (lines 493-503).

4. The data analysts obviously cannot be blinded to the data! How can they do comparisons and test the primary outcome if they are blinded? What needs to be blinded is the assessment of the primary outcome, which should be done independently of those providing the intervention.

Thank you for your valuable observation. We agree that it is not feasible to blind the data analysts to the data itself. To address your concern, we have clarified that the assessment of the primary outcome will be performed by an independent evaluator, who will be blinded to the group assignments (lines 507-512 and 568-569). This ensures that the outcome assessment is done objectively and independently from those providing the intervention. The data analysts will have access to the full dataset for the statistical analysis.

5. Very complicated set of interventions, but if successful, the results will be interesting.

Thank you for your thoughtful feedback. We appreciate your recognition of the complexity and potential impact of our study.

Reviewer #2: Manuscript PONE-D-24-31135

Title: Efficacy and Safety of the Early Implementation of a Multimodal Rehabilitation Program in Mechanically Ventilated Patients: A Randomized Clinical Trial Protocol

Thank you to the authors for a well-articulated and referenced manuscript. The study protocol requires additional methodological detail and consideration of the most appropriate outcome measure for this study. The manuscript will be publishable with moderate adjustments, particularly to the Introduction and Methods, in this reviewer’s view.

The fundamental question for the authors in the study design is to consider whether starting the input two days (48hrs) apart is enough to create a measurable study effect, and whether the sample groups are large enough to observe this effect. For instance, tailoring interventions to the RASS score could result in participants from both groups receiving similar and, or low volumes of therapy because heavy sedation or agitation excludes them for two days of their stay from receiving ‘active’ treatments.

Please adjust the manuscript and or respond to the comments below.

Abstract:

1. Please make it clearer for the reader as to how will the RASS score affect therapy input.

Thank you for your helpful feedback. We have revised the abstract to clarify how the RASS score influences therapy input (lines 55-62). Specifically, therapy intensity and type are now explicitly adjusted according to the patient’s RASS score, with passive interventions for those with lower scores (indicating deeper sedation) and more active therapies for patients with higher scores (indicating lighter sedation or alertness). This adaptive approach ensures that the therapy is tailored to each patient’s clinical condition (Table 1 and lines 224-236).

Introduction:

2. The study justification as articulated in the Introduction needs rework. The primary outcome measure is not post-ICU outcomes as purported as the main reason for early intervention, it is time on the ventilator. Why would the investigators not use a primary measure that is a post-ICU physical or quality of life outcome, as justified in by the Introduction?

Thank you for your valuable feedback. In response to your comment, we have revised the Introduction to clarify the justification for our study. Specifically, we have emphasized that the duration of mechanical ventilation is a key driver of adverse outcomes, such as those seen in post-intensive care syndrome (PICS) and is therefore an important target for early intervention. We have also highlighted how the primary outcome (time on the ventilator) aligns with the study’s aim of reducing long-term complications by addressing this critical factor (lines 76-86).

Methods:

3. Eligibility Criteria – requires additional detail and justification.

Thank you for your feedback regarding the eligibility criteria. We have revised this section to provide additional detail and justification for each inclusion and exclusion criterion (lines 183-190). Specifically, we now clarify that:

• A Barthel score ≥70 ensures participants have a reasonable level of functional independence pre-ICU, allowing for accurate assessment of the rehabilitation intervention’s impact.

• Invasive mechanical ventilation >24 hours ensures that patients have a sufficient duration of mechanical support to benefit from early rehabilitation.

4. Will patients with ICU length of stay shorter than 72 hours be included? How will this be accounted for in the study and statistical analyses? How might this impact the efficacy of group allocation (randomisation)?

Patients with ICU stays shorter than 72 hours will be included in the study. However, we recognize that shorter stays might impact their exposure to the full intervention. To address this:

• Primary Analysis: We will conduct an intention-to-treat analysis, where all randomized patients are analyzed in the groups they were assigned to, regardless of the length of their ICU stay or the level of intervention received.

• Secondary Analysis: We will perform a sensitivity analysis excluding patients with stays shorter than 72 hours to assess how this impacts the results. Additionally, a subgroup analysis will compare outcomes between patients with longer and shorter ICU stays.

• Statistical Adjustments: The duration of ICU stay will be included as a covariate in the statistical models to control for its impact on the primary outcome.

• Randomization Impact: Randomization will occur upon ICU admission, ensuring that the group assignment is not biased by predicted ICU stay. We will assess balance between groups post-randomization and implement adjustments if necessary.

(See statistical analysis section).

5. Line 181 - Will patients with isolated smoke inhalation injury (airway burn) be excluded? If so, why?

Thank you for your question. Yes, patients with isolated smoke inhalation injury (airway burn) will be excluded from the study (line 198). It is well known that these patients often require prolonged mechanical ventilation due to the nature of their injury, which could introduce significant variability into our study outcomes. Given that the primary outcome is the duration of mechanical ventilation, including these patients could skew the results and make it difficult to assess the effects of the intervention on the general ICU population (https://doi.org/10.1042/CS20040135).

6. Line 183 - Why are patients with COPD excluded?

Thank you for your question. Patients with chronic obstructive pulmonary disease (COPD) are excluded from the study because they tend to require prolonged mechanical ventilation due to their underlying respiratory dysfunction. Including COPD patients could introduce variability in the primary outcome (duration of mechanical ventilation), making it more challenging to isolate the effects of the intervention on the general ICU population.

According to Sellares et al. (2011), COPD was identified as a significant predictor of prolonged weaning from mechanical ventilation, with an odds ratio of 2.97 (95% CI: 1.29–6.82). This finding further supports the exclusion of COPD patients to ensure that the study results are not biased by this group’s typically prolonged ventilation requirements.

Reference:

1. Sellares J, Ferrer M, Cano E, et al. Predictors of prolonged weaning and survival during ventilator weaning in a respiratory ICU. Intensive Care Medicine. 2011;37(5):775-784.

7. Line 187 - How will life expectancy criteria be applied at recruitment?

Thank you for your question. The life expectancy criterion (≤180 days) will be applied at recruitment based on a combination of factors, including:

1. Primary diagnosis and comorbidities, such as advanced cancer, end-stage heart failure, or other critical irreversible conditions.

2. Medical judgment of the ICU team, considering the patient’s recent clinical course and overall prognosis.

3. Functional status, as measured by the Barthel index, which is already included in our study to assess the patient’s ability to perform activities of daily living.

This multi-faceted approach will ensure that only patients who are likely to benefit from the intervention are included in the study, while avoiding potential bias introduced by including patients with very limited life expectancy (see exclusion criteria).

8. Line 204 - Currently the daily number of treatments is not standardised. This reviewer strongly suggests that the number of intervention events be standardised to one or two per day per discipline to reduce study variability.

9. Similarly, the volume of treatment (minutes of input) applied per participant is a significant variable in this study. How will the investigators account for the daily differences?

10. Line 212 - Is there a better way to schedule the interventions to reduce variation in active treatment volume per patient? For instance, rather than accept that therapy is not completed or sessions markedly shortened due to periods of high sedation and agitation, could the investigators apply a RASS driven protocol which optimises the daily input (minimises down time) for the patient through the standardising treatment according to the strengths of different disciplines of the team?

Thank you for your insightful feedback. In response to your comments, we have standardized the interventions while maintaining different volumes for each therapy (Table 1). Speech therapy will consist of two sessions of 10 minutes per day, while both physical therapy and respiratory therapy will involve two sessions of 30 minutes per day. Occupational therapy will be provided once daily for 30 minutes. We have also aligned the interventions with a RASS-driven protocol to optimize treatment and minimize downtime.

• For patients with a RASS ≤ -2, passive and passive-assisted interventions will focus on maintaining joint mobility, sensory input, and pulmonary function.

• For patients with a RASS between -1 and +1, more active rehabilitation will be provided, including motor activation, respiratory exercises, and cognitive tasks.

• For patients with a RASS ≥ 2, physical, speech, and respiratory therapies will continue as prescribed for the -1 to +1 group. However, occupational therapy will shift focus towards behavioral and environmental modulation to reduce agitation, incorporating sensory stimuli and family education to promote spatiotemporal orientation.

11. Line 299 – How will adverse events be managed in the study analyses?

Adverse events will be systematically recorded and categorized according to their severity and their relationship to the intervention. Serious adverse events include altered blood pressure, cardiac arrhythmia, oxygen desaturation, agitation requiring sedation, removal of invasive lines, altered neurological status, accidental extubation, healthcare-associated pneumonia, and bronchoaspiration. These events will be carefully monitored throughout the study.

Adverse events will be analyzed descriptively, with the frequency and type of events reported for each study group. In addition, an intention-to-treat (ITT) analysis will be applied for the primary outcome, ensuring that participants who are withdrawn due to serious adverse events remain in the analysis according to their original group assignment. Secondary safety analyses will be conducted to compare the incidence of adverse events between the groups and to assess the overall safety profile of the interventions (see Patient withdrawal and statistical analysis sections)

12. Line 308 – Will the patient withdrawn due to complications no longer receive treatment

---

## [Decision Letter · Decision Letter 1]

21 Nov 2024

PONE-D-24-31135R1Efficacy and Safety of the Early Implementation of a Multimodal Rehabilitation Program in Mechanically Ventilated Patients: A Randomized Clinical Trial ProtocolPLOS ONE

Dear Dr. Alvarado Sánchez,

Thank you for submitting your manuscript to PLOS ONE. After careful consideration, we feel that it has merit but does not fully meet PLOS ONE’s publication criteria as it currently stands. Therefore, we invite you to submit a revised version of the manuscript that addresses the points raised during the review process.

We look forward to receiving your revised manuscript.

Kind regards,

Steven E. Wolf, MD

Academic Editor

PLOS ONE

Journal Requirements:

We note that your manuscript indicates that you plan to start data collection in October 2024. As studies presented in a Registered Report Protocol should not begin data collection before the submission has been accepted for publication, please confirm whether data collection has not yet started

Reviewers' comments:

Reviewer's Responses to Questions

**Comments to the Author**

1. Does the manuscript provide a valid rationale for the proposed study, with clearly identified and justified research questions?

Reviewer #1: Yes

Reviewer #2: Yes

2. Is the protocol technically sound and planned in a manner that will lead to a meaningful outcome and allow testing the stated hypotheses?

Reviewer #1: Yes

Reviewer #2: Partly

3. Is the methodology feasible and described in sufficient detail to allow the work to be replicable?

Reviewer #1: Yes

Reviewer #2: Yes

4. Have the authors described where all data underlying the findings will be made available when the study is complete?

Reviewer #1: Yes

Reviewer #2: Yes

5. Is the manuscript presented in an intelligible fashion and written in standard English?

Reviewer #1: Yes

Reviewer #2: Yes

6. Review Comments to the Author

You may also provide optional suggestions and comments to authors that they might find helpful in planning their study.

Reviewer #2: Thank you to the authors for their respectful and responsive revisions. There are additional minor clarifications requested to ensure that the study protocol would be replicable by others and for the authors to review exclusion criteria in order to prevent unnecessary exclusions and, or early terminations from the study (please address 31 comments in the attached PDF).

Lastly, please also reconsider the choice of primary outcome. This reviewer argues that the current plan is not reflective of the ethos of the study intervention and the authors be implored to be brave in their choice of a patient centric and allied health linked primary outcome.

7. PLOS authors have the option to publish the peer review history of their article (what does this mean?). If published, this will include your full peer review and any attached files.

Reviewer #1: No

Reviewer #2: **Yes: **Dale W. Edgar

---

## [Author Response · Author response to Decision Letter 2]

6 Jan 2025

Letter to reviewer

Thank you for allowing us to send you the revised version of our study protocol. Below, you will find the responses to each of your requirements.

1. Thank you for making these adjustments. It is important that the interventions are tailored to behavior eg agitation, not simply a RASS score as patient alertness does not equate directly with cooperation with therapy.

Response: Thank you for your thoughtful observation. We recognize that tailoring interventions to patient behavior, such as agitation or cooperation, is crucial. As detailed in the RASS-Driven Protocol (lines 248-258) the RASS score serves as a foundational guide, but therapy interventions are further adjusted in real-time based on the patient’s behavior and readiness to cooperate. This ensures a personalized approach that considers both sedation levels and the patient’s behavioral state.

To enhance clarity, we have updated the abstract to reflect this nuance, ensuring consistency with the protocol and manuscript (lines 61-64). We appreciate your attention to this important detail.

2. This response is remarkably challengeable as much as it relates to the choice of outcome which is convenient.

Is mechanical ventilation a key driver of adverse outcomes or is it a marker of other factors which are far more likely to correlate with complications and adverse outcomes, especially post-ICU syndrome? For example, to be mechanically ventilated requires sedation and muscle relaxants as well as a cocktail of other pharmacological interventions which are likely to be primary drivers of post-ICU syndrome. Further, bed rest associated with mechanical ventilation is likely to be implicated as a direct cause of multiple adverse events.

Is there a specific reason that the authors are avoiding a more contemporary approach with a patient centric measure, particularly one that would be a marker of impact or severity of post-ICU syndrome, as their primary outcome?

Response:

Thank you for your insightful feedback. We acknowledge the nuanced relationship between mechanical ventilation and adverse outcomes such as post-ICU syndrome (PICS). While mechanical ventilation itself may not be the sole driver of these complications, it is a significant surrogate marker of critical illness severity and a key modifiable factor in ICU management. As such, its duration is a clinically relevant outcome that aligns with our study's aim to mitigate long-term complications through early multimodal rehabilitation.

We have clarified in the manuscript that mechanical ventilation duration reflects not only the time spent on ventilatory support but also the cumulative effects of associated interventions, such as sedation, neuromuscular blockade, and immobility, all of which contribute to PICS. By targeting a reduction in ventilator days, we aim to address these underlying factors, thereby indirectly influencing post-ICU outcomes.

The choice of this primary outcome also considers practical and methodological reasons. Mechanical ventilation duration is:

• Objective and measurable: Unlike quality-of-life measures, which are often subjective and influenced by a range of confounding factors, ventilator duration is directly linked to ICU management strategies and provides a clear endpoint for evaluating the efficacy of early rehabilitation.

• Proven relevance: Studies, including those cited in the manuscript, have demonstrated that reducing ventilator days is associated with improved muscle strength, earlier mobilization, and potentially better long-term recovery.

We recognize the value of more patient-centric outcomes, particularly those that assess the severity or impact of PICS. To address this, we have included secondary and exploratory outcomes, such as post-extubation muscle strength, Barthel Index scores, and quality-of-life assessments at 28 and 90 days post-intubation. These outcomes will provide a broader perspective on the long-term impact of our intervention while complementing the primary outcome.

We have revised the Introduction to explicitly address these points, emphasizing the rationale for our primary outcome while acknowledging its relationship to other factors driving adverse outcomes. Additionally, we highlight the inclusion of patient-centric measures as secondary endpoints to capture the broader impact of early rehabilitation.

We appreciate your thoughtful comment, which has helped refine and strengthen the study’s justification

3. This response is self contradicting in the context of the intervention. Would the authors not be interested in determining if their intervention was effective in reducing mechanical ventilation time for those with prolonged critical care support needs?

Response: Thank you for this insightful comment. We recognize the potential value of evaluating the intervention in patients with prolonged critical care support needs, such as those with isolated smoke inhalation injuries. However, our decision to exclude these patients was based on several considerations.

First, in our clinical setting, patients with airway burns often have concurrent extensive burns involving large areas of body surface. This adds significant complexity to their clinical course, including increased risk of infections, fluid imbalances, and metabolic demands, which further prolong mechanical ventilation and ICU stays. These factors introduce considerable variability that could confound the study outcomes and obscure the true effects of the multimodal rehabilitation program in the broader ICU population.

Second, the pathophysiology and clinical management of patients with airway burns differ substantially from those without these injuries. The severe airway inflammation, necrosis, and obstruction typical in these cases often require prolonged ventilatory support, independent of rehabilitation interventions. Including such a group might bias the results and make it challenging to assess the efficacy of our intervention for the general ICU population.

Finally, our primary goal in this trial is to evaluate the intervention in a more homogenous population where factors influencing the duration of mechanical ventilation—such as immobility and sedation practices—are modifiable through early rehabilitation. Including highly complex subpopulations like these could dilute the observed effects of the intervention.

That said, we acknowledge the importance of assessing the efficacy of our program in subpopulations with unique critical care needs. The findings from this study will help establish a foundation for future research to explore the program’s potential benefits in patients with airway burns and other complex conditions.

4. Thank you - please add a short justification with this reference to the Exclusion criteria.

Response: The justification for excluding COPD patients has been added to the exclusion criteria as requested (lines 215-217).

5. Thank you - please summarize this additional detail into the Exclusion criteria section.

Response: Thank you for your comment. We have summarized the additional details regarding the life expectancy criterion into the exclusion criteria section, specifying that patients with a life expectancy of ≤180 days will be excluded based on their primary diagnosis, comorbidities, recent clinical course, medical judgment by the ICU team, and functional status as assessed by the Barthel index (lines 222-225).

6. Could the Clavien-Dindo classifications be an appropriate framework to use in this instance?

Response: Thank you for your comment. The Clavien-Dindo classification has been exclusively used to categorize surgical complications. Thus, this classification is inappropriate for our study. We decided to use the Common Terminology Criteria for Adverse Events classification which has been used primarily in oncological studies, but it is also used in other context such as clinical trials (lines 270-302).

7. This is important however, if the primary outcome remains time on the ventilator, how will the ITT principle be able to be applied if the patient is withdrawn while still ventilated?

Also, if the adverse event is deemed by the treating intensive care physician NOT to be related to the multimodal rehab, why would the patient be removed from the study and an adverse reaction recorded as associated with the study intervention?

Response: Thank you for your insightful question. The intention-to-treat (ITT) principle will still be applied to ensure the integrity of the primary outcome, even for patients who are withdrawn while still ventilated. Specifically:

a) Primary Outcome (Time on Ventilator): For patients withdrawn due to complications, their time on the ventilator will be recorded up until the point of withdrawal. This data will be included in the analysis as censored observations, acknowledging that the true duration of ventilation could not be fully captured due to withdrawal. This approach ensures that their contribution to the primary outcome is preserved without biasing the results.

b) Statistical Handling: In the analysis, survival methods such as Kaplan-Meier estimates and Cox regression models will be used to account for censored data. This ensures that the duration of ventilation for withdrawn patients is appropriately incorporated without compromising the ITT principle or the validity of the results.

c) Subgroup and Sensitivity Analyses: Additional analyses will explore the potential impact of withdrawn patients on the study outcomes. These analyses will help determine whether their withdrawal introduces any systematic differences and ensure the robustness of the study conclusions.

Not all participants who experience an adverse event or complication will be withdrawn from the study. In the literature, most common adverse events in clinical trials are grade 1 and 2. Therefore, participants experiencing minor adverse events should not be withdrawn from the study solely for experiencing an adverse event, but based on an analysis of the adverse event. Nonetheless they will be informed regarding the adverse event and ask if they want to continue to participate. Severe adverse events will be analyzed by the ethics committee and they will determine if the participant continues in the study (lines 292-307).

By combining these methods, we aim to uphold the ITT principle while balancing patient safety and maintaining the validity of the primary outcome analysis. Thank you for raising this critical point. It has helped clarify our approach to managing these cases.

8. Thank you for these changes. There are multiple non-patient related reasons for why a participant does not adhere to a research study protocol - including staffing in the ICU; surgery; procedures including central line changes and insertions; and, poorly controlled sedation practices. To exclude patients from the study for reasons such as these may impact the study sample size greatly and thus, it would be an enhancement to the design if it was adjusted to maintain patients with deviations from protocol which are related to such events which are beyond the control of the patient and the researchers.

Response: Thank you for your thoughtful feedback. We completely agree that deviations from the protocol due to factors beyond the control of the patient or researchers, such as ICU staffing limitations, surgeries, procedures, or sedation management, should not lead to automatic withdrawal. To address this concern, we have revised our protocol to account for these scenarios and ensure the study sample size is maintained without compromising the integrity of the analysis (lines 381-411).

9. Is there any reason why these data couldn't be recorded daily? For example, with the morning physiotherapy treatment, record the ventilator settings.

Response: Thank you for your thoughtful suggestion. We agree that collecting ventilator parameters daily, particularly in conjunction with the morning physiotherapy treatment, could provide valuable insights into the dynamic evolution of mechanical ventilation and its relationship with the intervention.

To enhance the study design while balancing feasibility, we propose the following adjustments:

a) Daily Collection: Ventilator parameters will be recorded daily, at a consistent time, ideally before the physiotherapy session. This will ensure that data are standardized and reflective of the patient's baseline respiratory status before the intervention (lines 426-429).

b) Expanded Analysis: These additional data points will be analyzed as an exploratory outcome to assess trends in ventilator support requirements over time and their relationship to the intervention. This could provide further insights into the impact of the program on ventilatory dependency (lines 490-506).

c) Primary Outcome Integrity: The daily recordings will complement the primary outcome analysis without introducing additional complexity, as the primary comparison of baseline ventilator settings will still focus on demonstrating equivalence between groups.

By implementing this adjustment, we aim to enrich the study data without overburdening the clinical team. Thank you for highlighting this opportunity to strengthen our research methodology.

10. Please add these explanations into the Statistical Analysis methods.

Response: We have incorporated these explanations into the Statistical Analysis methods section for clarity and transparency (lines 654-733).

11. Reference please.

Response: We added a reference in the introduction (Na SJ, Ko R-E, Nam J, Ko MG, Jeon K. Factors associated with prolonged weaning from mechanical ventilation in medical patients. Ther Adv Respir Dis. 2022;16. doi:10.1177/17534666221117005).

12. Reference please. As noted, mechanical ventilation is a convenient indicator for statistical purposes, but is it the best measure to identify the impact of an early MRP?

Response: Thank you for your observation. Mechanical ventilation was chosen as an indicator due to its well-documented utility in critically ill patients as a proxy for disease severity and the progression of organ dysfunction. While we acknowledge that mechanical ventilation may not be the sole or perfect measure to evaluate the impact of an early MRP, it provides a quantifiable, objective, and clinically relevant endpoint. Moreover, its association with ICU outcomes makes it a convenient and practical indicator for statistical analyses in this context.

13. Is this exclusion applied if the patient has had H&N surgery at any time in the past or within last 2 years prior to admission? How will acute H&N grafting surgery during the ICU admission affect the study participation? Will the patient be excluded mid-protocol?

Response: Thank you for your thoughtful question. In response, we have refined the exclusion criterion and clarified our approach to managing cases involving head and neck (H&N) surgeries:

• Exclusion Criterion: Patients with a history of head and neck surgery at any time prior to ICU admission will be excluded from the study. This is based on the well-documented risk of dysphagia and/or bronchoaspiration associated with head and neck surgeries, which could confound the interpretation of our results by introducing variables unrelated to the MRP being evaluated (line 211).

• Management of Acute Head and Neck Surgeries During ICU Stay:

Patients who undergo acute head and neck surgery during their ICU stay will not be withdrawn from the study. Instead, these cases will be managed as handling minor deviations/external factors:

o Deviations due to such surgeries, along with other ICU procedures (e.g., central line insertions), staffing limitations, or uncontrolled sedation practices, will not lead to participant withdrawal.

o These events will be documented as protocol deviations and addressed in sensitivity analyses to evaluate their potential impact on study outcomes.

This approach ensures adherence to the intention-to-treat principle while maintaining the study’s methodological integrity. By excluding patients with a prior history of head and neck surgery and managing intra-study surgeries as deviations, we aim to minimize confounding variables and ensure a robust analysis of the MRP’s effects.

14. Is this exclusion app

---

## [Decision Letter · Decision Letter 2]

24 Apr 2025

Efficacy and Safety of the Early Implementation of a Multimodal Rehabilitation Program in Mechanically Ventilated Patients: A Randomized Clinical Trial Protocol

PONE-D-24-31135R2

Dear Dr. Alvarado Sánchez,

We’re pleased to inform you that your manuscript has been judged scientifically suitable for publication and will be formally accepted for publication once it meets all outstanding technical requirements.

Kind regards,

Luis Felipe Reyes, M.D., Ph.D., MSc.

Academic Editor

PLOS ONE

Reviewers' comments:

Reviewer's Responses to Questions

**Comments to the Author**

1. Does the manuscript provide a valid rationale for the proposed study, with clearly identified and justified research questions?

Reviewer #1: Yes

Reviewer #2: Yes

2. Is the protocol technically sound and planned in a manner that will lead to a meaningful outcome and allow testing the stated hypotheses?

Reviewer #1: Yes

Reviewer #2: Yes

3. Is the methodology feasible and described in sufficient detail to allow the work to be replicable?

Reviewer #1: Yes

Reviewer #2: Yes

4. Have the authors described where all data underlying the findings will be made available when the study is complete?

Reviewer #1: Yes

Reviewer #2: Yes

5. Is the manuscript presented in an intelligible fashion and written in standard English?

Reviewer #1: Yes

Reviewer #2: Yes

6. Review Comments to the Author

You may also provide optional suggestions and comments to authors that they might find helpful in planning their study.

Reviewer #1: All my comments have been addressed adequately. xxxxxxxxxxxxxxxxxxxxxxxxxxxxxxxxxxxxxxxxxxxxxxxxxxxx

Reviewer #2: Manuscript PONE-D-24-31135_R2

Title: Efficacy and Safety of the Early Implementation of a Multimodal Rehabilitation Program

in Mechanically Ventilated Patients: A Randomized Clinical Trial Protocol

Thank you to the authors for their considered and well-articulated responses to the latest review. The additional scientific justifications, increased frequency of primary outcome collection and extra secondary measures improve the study methods replicability as well as the likelihood of achieving interpretable outcomes, if this study is conducted as per protocol. The increased sophistication and clarity of the responses with subsequent reviews demonstrates that the authors have improved this trial and their own research knowledge and skills remarkably during this review process. Thank you to the authors for engaging in the peer review process wholeheartedly and developing this study.

Please note that Ethics Committees are not at the beck and call of researchers and to ask the EC to make real-time determinations of withdrawal of patients from a study in relation to major protocol deviations (added text, Lines 398-405) is not likely to be implementable and could expose the patient to unsafe study conditions. In preference, consider identifying a local clinician such as a ICU physician, preferably independent to the study who can be consulted to make the decision to remove the patient from the study due to adverse events and, or protocol deviations.

In relation to the generalisability and replication of this protocol elsewhere, the author’s points in responses 18 and 26/27 raise one last question. ICU and mechanical ventilation practices for acute burn patients vary considerably around the world. In that, conducting a trial such as this would not be feasible in the reviewer’s burn service because very few patients each year, have an ICU length of stay greater than three days. Do the authors note that only 8 ventilated burn patients were admitted, in total, to their service in 2023 and thus, is this study feasible in the author’s burn unit if it is likely to take > 7 years to complete? It seems an important point to make in this manuscript, that a multisite, collaborative approach is advocated for in order to complete this study in a viable timeframe. Please consider adding that point into Methods.

7. PLOS authors have the option to publish the peer review history of their article (what does this mean?). If published, this will include your full peer review and any attached files.

Reviewer #1: No

Reviewer #2: **Yes: **Dale W Edgar

---

## [Editor Report · Acceptance letter]

PONE-D-24-31135R2

PLOS ONE

Dear Dr. Alvarado Sánchez,

I'm pleased to inform you that your manuscript has been deemed suitable for publication in PLOS ONE. Congratulations! Your manuscript is now being handed over to our production team.

Kind regards,

on behalf of

Dr. Luis Felipe Reyes

Academic Editor

PLOS ONE